# Understanding variations in downwelling longwave radiation using Brutsaert's equation

Yinglin Tian[1,2], Deyu Zhong[1], Sarosh Alam Ghausi[2,3], Guangqian Wang[1], Axel Kleidon[2]

[1]State Key Laboratory of Hydroscience and Engineering, Department of Hydraulic Engineering, Tsinghua University, 100084 Beijing, China.
[2]Biospheric Theory and Modelling, Max Planck Institute for Biogeochemistry, 07701 Jena, Germany
[3]International Max Planck Research School on Global Biogeochemical Cycles (IMPRS-gBGC), 07701 Jena, Germany

*Correspondence to*: Axel Kleidon (akleidon@bgc-jena.mpg.de)

## Abstract

A dominant term in the surface energy balance and central to global warming is downwelling longwave radiation ($R_{ld}$). It is influenced by radiative properties of the atmospheric column, in particular by greenhouse gases, water vapour, clouds and differences in atmospheric heat storage. We use the semi-empirical equation derived by Brutsaert (1975) to identify the leading terms responsible for the spatial-temporal climatological variations in $R_{ld}$. This equation requires only near-surface observations of air temperature and humidity. We first evaluated this equation and its extension by Crawford and Duchon (1999) with observations from FLUXNET, the NASA-CERES dataset, and the ERA5 reanalysis. We found a strong spatiotemporal correlation between estimated $R_{ld}$ and the datasets above, with $r^2$ ranging from 0.87 to 0.98 across the datasets for clear-sky and all-sky conditions. We then used the equations to show that changes in lower atmospheric heat storage explain more than 95% and around 73% of diurnal range and seasonal variations in $R_{ld}$, respectively, with the regional contribution decreasing with latitude. Seasonal changes in the emissivity of the atmosphere play a second role, which is controlled by anomalies in cloud cover at high latitudes but dominated by water vapor changes at mid-latitudes and subtropics, especially over monsoon regions. We also found that as aridity increases over the region, the contributions from changes in emissivity and lower atmospheric heat storage tend to offset each other (-40 W m$^{-2}$ and 20-30 W m$^{-2}$, respectively), explaining the relatively small decrease in $R_{ld}$ with aridity (-(10-20) W/m$^{-2}$). These equations thus provide a solid physical basis for understanding the spatiotemporal variability of surface downwelling longwave radiation. This should help to better understand and interpret climatological changes, such as those associated with extreme events and global warming.

## 1 Introduction

In the global mean surface energy budget, downward longwave radiation ($R_{ld}$) is dominant surface energy input (333 W/m$^2$ in global mean and 306 W$^2$/m over land), contributing around twice as much energy as absorbed solar radiation (161 W/m$^2$ in global mean and 184 W$^2$/m over land) (Trenberth et al. 2009, Wild et al. 2015). This dominance holds over all regions in the climatological mean, although there are some clear variations in space and time (Figs. 1 and S1). It is central to global warming, reflecting the greenhouse effect of the atmosphere (Held and Soden 2000), and its variations have been suggested to be the main contributor to some regional warming amplifications, such as in the Arctic (Lee et al. 2017) and the Tibetan Plateau (Su et al. 2017). Therefore, it is important to understand the main sources of variations in this surface energy balance term, which can be seen in Figure 1.

The flux of downwelling longwave radiation is influenced by the radiative properties of the entire atmospheric column, i.e., water vapour, clouds, and greenhouse gases, but also by the heat stored in the atmosphere, i.e., the temperature at which radiation is emitted back to the surface. To obtain an estimate of this flux, Brutsaert (1975) used functional expressions for the typical temperature and humidity profiles of the lower troposphere together with radiative transfer equations and semiempirical relationships of the absorptivity by water vapor, integrated these vertically, and expressed the resulting flux $R_{ld}$ in terms of near-surface air temperature and water vapour pressure for clear-sky conditions. He thereby derived a semi-empirical equation for $R_{ld}$ for an effective clear sky emissivity ($\varepsilon_{cs}$) and the corresponding flux of downwelling longwave radiation ($R_{ld,cs}$):

$$\varepsilon_{cs} = 1.24(e_a/T_a)^{1/7}, \tag{1}$$

$$R_{ld,cs} = \varepsilon_{cs}\sigma T_a{}^4. \tag{2}$$

where $\sigma$ is Stefan–Boltzmann constant ($\sigma$ = 5.67 10$^{-8}$ W m$^{-2}$ K$^{-4}$), $e_a$ is the 2m water vapor pressure (unit: millibars) and $T_a$ is the 2m air temperature (unit: K). The latter two meteorological variables can easily be obtained or inferred from weather stations, so that the downwelling flux of longwave radiation can be estimated from weather station observations. Note that the $\varepsilon_{cs}$ shown in equation 1 is largely insensitive to changes in $T_a$. As a result, emissivity does not have a direct dependence on T$_a$, except that higher temperature may also lead to higher values in e$_a$.

This equation was later extended to all-sky conditions that include the effects of cloud cover, among which Crawford and Duchon (1999) is a common extension (Alados et al. 2012; Duarte et al. 2006; Flerchinger et al. 2009). This extension diagnoses cloud cover fraction ($f_c$) as the fraction of incoming solar radiation at the surface ($R_s$) in relation to the potential solar radiation ($R_{s,pot}$), that is, the incoming flux at the top of the atmosphere. The emissivity for all-sky conditions, $\varepsilon$, is then calculated as the mix of the emissivities of clear-sky conditions (Eqn. (1), weighted by the cloud-free proportion, ($1 - f_c$) and clouds with an emissivity of $\varepsilon_c = 1$ (weighted by the cloud fraction $f_c$). Using this emissivity, the estimation of downwelling longwave radiation is then done by

$$f_c = 1 - R_s/R_{s,pot}, \tag{3}$$

$$\varepsilon = f_c + (1 - f_c)\varepsilon_{cs}, \tag{4}$$

$$R_{ld} = \varepsilon\sigma T_a{}^4. \tag{5}$$

Previous studies have already verified Equations 4-5 to have a very good agreement with site measurements with the r$^2$ of 0.883 and RMSE of 15.367 W/m$^2$ (Duarte et al. 2006; Hatfield et al. 1983), especially when the temperature is higher than 0℃ (Aase and Idso 1978; Satterlund 1979). Other studies have worked to calibrate and modify this estimate further to different regions (Malek 1997; Sridhar and Elliott 2002).

This expression for downwelling longwave radiation $R_{ld}$ given by Eqn. (5) allows us to quantify the different contributions by cloud cover, $f_c$, water vapor concentrations, $e_a$ (as a measure of the total water vapor content

of the atmospheric column), and air temperature, $T_a$ (as a proxy for the heat storage within the lower
atmosphere, Panwar et al. 2022). With this, we can then attribute variations in $R_{ld}$ to their physical causes.
Here, our aim is to first evaluate this estimate for downwelling longwave radiation with current global
datasets at the continental scale. These variations are illustrated using the NASA-CERES (EBAF 4.1)
dataset (Loeb et al., 2018; Kato et al., 2018, NASA/LARC/SD/ASDC 2017) and the NASA-CERES
Syn1deg dataset (Doelling et al., 2013, 2016) in Figure 1 and are compared to variations in solar radiation.
It can be seen that the climatological distribution of $R_{ld}$ is mostly associated with latitudes, while also
presenting some zonal variations, e.g., across western and eastern North America. In comparison, the
seasonal cycle of $R_{ld}$ is less determined by latitudes (Fig. 1b). It has a larger magnitude over land than over
oceans, over arid regions than humid regions, and over cold regions more than over warm ones. Although
studies have revealed a close correlation between the variation of $R_{ld}$ and other factors like air temperature,
water vapor, and $CO_2$ concentration (Wang and Liang 2009; Wei et al. 2021), here we go beyond
correlations and rather attribute these variations to the different terms in Eqns. (1)-(5) that represent
different radiative properties affecting $R_{ld}$.
To figure out the dominant driver for these spatiotemporal variations, we decompose changes in $R_{ld}$ into its
components: cloud cover, $f_c$, heat storage changes of atmosphere as reflected by 2m air temperature, $T_a$,
and air humidity, $e_a$, by performing the differentiation of these equations. We show that heat storage
changes predominantly shape the diurnal range and seasonal cycle of $R_{ld}$, while cloud cover variations play
a second role in most cases. In addition, the temporal variations of $R_{ld}$ are less over the ocean than over
land, and less during winter than summer. On the other hand, the spatial variations of $R_{ld}$ from arid to humid
regions is relatively small, which we will show is due to a compensating effect of corresponding changes
in atmospheric emissivity and heat storage.
Our paper is organized as follows: After briefly describing the datasets used in our evaluation in Section 2,
we first the estimate of $R_{ld}$ from these equations at the global scale, using multiple datasets in Section 3.1.
After showing that the annual-mean and large-scale variations are well captured, we then use the equations
to decompose the temporal variations of $R_{ld}$ in terms of its mean spatial and temporal variations and relate
these to their causes in Section 3.2. The spatial variations of $R_{ld}$ are then further discussed in Section 3.3 in
terms of its relationship with aridity. We then close with a brief summary and broader implications.

## 2 Datasets

To test $R_{ld}$ estimates, we use FLUXNET half-hour observations (Pastorello et al. 2020, half-hourly values,
189 sites, see Table S1 and Figure S2 for details), the NASA-CERES monthly satellite-based radiation
dataset (Doelling et al., 2013, 2016, monthly means, covering years 2001 to 2018), and the ERA5 monthly
reanalysis dataset (Hersbach et al. 2018, monthly means, covering years 1979 to 2021).
For each dataset, $T_a$, $e_a$, and $f_c$ are needed as inputs for Eqs. (1)-(5), while $R_{ld}$ data is used for the
comparison. Cloud cover $f_c$ is calculated using Eq. (3) for all three datasets with incoming solar radiation
at the surface ($R_s$) and the potential solar radiation ($R_{s,pot}$). For NASA-CERES estimation, $T_a$ from the
CPC Global Unified Temperature dataset (CPC Global Unified Temperature) is used as temperature
observation.
For all three datasets, water vapor pressure, $e_a$, is not directly given. It is calculated from the water vapor
deficit (VPD, FLUXNET) or dewpoint temperature ($T_{dew}$, ERA5) using Monteith and Unsworth (2008):

$$e_a = 6.1079 \times \exp\left(17.269 T_{dew}/(237.3 + T_{dew})\right), \tag{6}$$

$$e_a = 6.1079 \times \exp\left(17.269 T_a/(237.3 + T_a)\right) - VPD, \tag{7}$$

And the calculated $e_a$ from ERA5 is also used in NASA-CERES estimation.
For the analysis of the spatial variations of $R_{ld}$ along water availability, we use the aridity index (AI $= \frac{R}{LP}$)
(Budyko 1958; UNCOD 1977). This index is calculated using the mean annual net radiation ($R$) taken from
the NASA-CERES dataset, the mean annual net precipitation ($P$) taken from the CPC Global Unified
Gauge-Based Analysis of Daily Precipitation data (Chen et al. 2008 and Xie et al. 2007, CPC Global Unified
Gauge-Based Analysis of Daily Precipitation), and a latent heat of vaporization for water of $L =$
2260 kJ/kg. A larger value of AI indicates stronger aridity.

## 3 Results and discussion


### 3.1 Comparison to observed, satellite, and reanalysis data


We first compared the estimates of $R_{ld}$ at a point-by-point basis separately for clear-sky and all-sky
conditions using Eqns. (2) and (5), respectively. This comparison is shown in Figure 2 using FLUXNET,
CERES, and ERA5 data. The estimates correlate very well with $r^2$ of 0.92 and 0.87 for clear-sky and all-
sky conditions, respectively, and RMSE values of 18.24 and 24.56 W m$^{-2}$. The slope of the linear
regressions between the estimated and observed $R_{ld}$ for FLUXNET are 1.03 and 1.02, with most data points
being concentrated around the 1:1 line (Figs. 2a and 2b). Note that for all-sky conditions, the agreement is
slighty less good, with a lower correlation coefficient and a larger RSME. The agreement with the NASA-
CERES and ERA5 datasets are even better, with higher correlation coefficients and lower RSME.
Despite this high level of agreement of the estimates, we can see some systematic biases in the estimates
for $R_{ld}$. These can be seen in Figure 3 and Figure S3, which show the spatial distribution of these biases
and their variations against temperature and humidity. For clear-sky conditions, there appears to be a
general underestimation in the high latitudes and, to some extent, in arid regions (Figs. 3c and 3e). Brutsaert
(1975) already described that for very low temperatures and in arid conditions, there are better parameter
values than those used in Eq. 1, with a larger coefficient than 1.24 and a different exponent. This can then
lead to an underestimation of Rld under low humidity (Figs. 3a, S3a, S3c). Moreover, B75 has not
considered the gradual increase in emissivity as temperature decreases below freezing (Aase and Idso
1978), thus explaining the underestimation under low temperature (Figs. 3b, S3b, S3b). The biases seen in
Figure 3 are nevertheless notably smaller than the spatial-temporal variations shown in Figure 1. This means
that these biases do not prevent us from using Brutsaert to attribute the causes for the seasonal variation
and the spatial range of R$_{ld}$.
The biases for all-sky conditions generally share the distribution with that of clear-sky conditions, with a
smaller magnitude (Figs. 3b, 3d and 3f), which are also small compared to the spatial-temporal variations.
Overall, this evaluation shows that the expressions given by Eqns. (1) - (5) are very well suited to describe
the spatiotemporal variations of $R_{ld}$ for current climatological conditions.

### 3.2 Attribution of diurnal and seasonal variations


We next use Eqns. (1) - (5) to attribute temporal variations of $R_{ld}$ to their physical causes. To do so, we can
express changes $\Delta R_{ld}$ as a function of changes in water vapor, $\Delta e_a$, cloud cover, $\Delta f_c$, and air temperature,
$\Delta T_a$. The functional dependence is derived from the equations by differentiation and applying the chain
rule. In a first step, we express a change $\Delta R_{ld}$ by the partial contributions $\Delta R_{ld,\varepsilon}$ and $\Delta R_{ld,T}$, that are due to
changes in emissivity, $\Delta \varepsilon$, and due to changes in atmospheric heat storage that are associated with a change
in air temperature $\Delta T_a$:

$$\Delta R_{ld} = \Delta R_{ld,\varepsilon} + \Delta R_{ld,T} = \frac{\partial R_{l,d}}{\partial \varepsilon}\Delta \varepsilon + \frac{\partial R_{l,d}}{\partial T_a}\Delta T_a = \sigma \overline{T_a}^4 \Delta \varepsilon + 4\sigma \bar{\varepsilon}\overline{T_a}^3 \Delta T_a. \tag{8}$$

The 2 terms at the right side of Eq. 8 are $\Delta R_{ld,\varepsilon}$ and $\Delta R_{ld,T}$, respectively.
The contribution $\Delta R_{ld,\varepsilon}$ is further decomposed into contributions $\Delta R_{ld,f_c}$, $\Delta R_{ld,e_a}$, and $\Delta R_{ld,T_a}{}'$ due to
variations in clouds, $\Delta f_c$, air humidity, $\Delta e_a$, and surface temperature, $\Delta T_a$. We obtain:

$$
\begin{aligned}
\Delta R_{ld,\varepsilon} = \sigma\,\overline{T_a}^4 \Delta\varepsilon \approx{} & \sigma\overline{T_a}^4 \times \frac{\partial\varepsilon}{\partial f_c}\Delta f_c + \sigma\overline{T_a}^4 \times \frac{\partial\varepsilon}{\partial e_a}\Delta e_a + \sigma\overline{T_a}^4 \times \frac{\partial\varepsilon}{\partial T_a}\Delta T_a \\
={} & \sigma\overline{T_a}^4 \times \left(1 - \overline{1.24\left(\frac{\overline{e_a}}{\overline{T_a}}\right)^{\frac{1}{7}}}\right)\Delta f_c + \sigma\overline{T_a}^4 \times \frac{1.24}{7}\frac{\left(1-\overline{f_c}\right)}{(\overline{e_a})^{\frac{6}{7}}(\overline{T_a})^{\frac{1}{7}}}\Delta e_a \\
& + \sigma\overline{T_a}^4 \times \left(-\frac{1.24}{7}\right) \times \frac{\left(1-\overline{f_c}\right)(\overline{e_a})^{\frac{1}{7}}}{(\overline{T_a})^{\frac{8}{7}}} \times \Delta T_a) \,.
\end{aligned}
\qquad (9)
$$

The 3 terms at the right side of Eq. 9 are $\Delta R_{ld,f_c}$, $\Delta R_{ld,e_a}$, and $\Delta R_{ld,T_a}{}'$, respectively.
Note that the third term is of less magnitude compared with the other two terms (e.g. in terms of the seasonal
range as shown in Fig. 5f), which is hence not focused in this work.
We next applied this approach to the diurnal deviations $\Delta R_{ld}$ from the daily mean using the FLUXNET
dataset. This decomposition is shown in Figure 4 in aggregated form across the FLUXNET sites for whole
year (Fig. 4a), the Northern hemisphere summer (Fig. 4b) and winter seasons (Fig. 4c). More than 95% of
the diurnal variations (of about $\pm$ 20 W m$^{-2}$) are caused by diurnal changes in air temperature, while
variations in emissivity play practically no role (Fig. S4). Diurnal changes in air temperature reflect
variations in heat storage of the atmospheric boundary layer. This is consistent with the notion that diurnal
variations in solar radiation over land are buffered primarily by the lower atmosphere, rather than below
the surface as it is the case for open water bodies and the ocean (Kleidon and Renner 2017). Since most of
the stations in the FLUXNET dataset are located in the midlatitudes of the Northern hemisphere, the
variations are consistently larger in summer due to the greater solar input (Fig. 4b) than in winter (Fig. 4c).
Figure 5 shows the same kind of decomposition, but for seasonal variations in $R_{ld}$ in the NASA-CERES
dataset, which is the difference between the maximum and minimum of monthly $R_{ld}$ data. Generally, areas
with relatively low annual-mean $R_{ld}$, e.g. the high latitude regions of North America and northeastern
Eurasia, have the largest seasonal cycle (Fig. 1). The decomposition shows that this variation is mostly due
to the seasonal variation in atmospheric heat storage ($\Delta R_{ld,T}$), with a portion of around 73% on a global
scale, and the rest are attributed to the seasonal changes in water vapor (24%) and cloud cover (12%).
Notably, seasonal variations in emissivity play a greater role than atmospheric heat storage in changing $R_{ld}$
in tropical areas, especially over the monsoon region. This is predominantly due to the strong seasonal
fluctuations in water vapor levels and cloud-cover (Figs. 5d-5f).
The aggregation to the global scale across land and ocean is shown in Fig. S5, where the deviations are
calculated as the difference of the monthly means to the annual mean. Figs. S5 show that the seasonal
variations of $R_{ld}$ is generally less over the ocean than on the land, an effect that can also be seen in Fig. 1.
The decomposition shows that these variations are mostly caused by changes in lower atmospheric heat
storage, with a slight modulation by emissivity changes. This can, again, be largely explained by the effect
described above for the diurnal variations (Kleidon and Renner 2017). Over the land, the changes in
radiation are majorly buffered by the heat storage in the lower atmosphere by the variations in convective
boundary layer height. However, over marine areas, solar radiation penetrates the transparent water bodies,
the heat storage of which hence buffers the season cycle of the radiation over the ocean. Since the heat
storage of the water body is larger than that of the lower atmospheric boundary layer, the buffering effect
is consequently larger, which leads to the less seasonal cycle of the surface temperature and R$_{ld}$ over the
ocean.
In summary, what our decomposition shows is that most temporal variations in $R_{ld}$ in current, climatological
conditions are explained by heat storage changes within the lower atmosphere.

### 3.3 Attribution of geographic variations with aridity

Last, we applied the decomposition to the climatological variations in $R_{ld}$ along with differences in mean water availability. Water availability was characterized by Budyko's aridity index (AI), with values AI < 1 representing humid regions, and larger values reflecting increased aridity. The spatial distribution of AI is shown in Fig. S6. Here, the deviations $\Delta R_{ld}$ are calculated with respect to the annual mean over land. The different contributions to the deviations are shown in Fig. 6, as well as the delineation along the aridity index (Figs. 6e - f).

The decomposition of the spatial distribution of the climatological means shows that the variations are largely caused by differences in lower atmospheric heat storage as well (Fig. 6a). The contribution due to variations in emissivity has a smaller magnitude (Fig. 6b), and is dominated by changes in cloud cover (Fig. 6c) and changes in water vapor (Fig. 6d) at high- and mid- latitudes respectively.

These variations are evaluated with respect to the aridity index in Figs. 6e, 6f and S7. While there is a large spread, as seen in the quantiles, there is a small, but consistent trend towards lower values of $R_{ld}$ in more arid regions, with a magnitude of about $-10{\sim}20$ W m$^{-2}$ across the entire aridity index spectrum (black dashed line in Figs. 6e and 6f). We also notice a shift in the contributions, with emissivity contributing less and lower atmospheric heat storage contributing more with increased values of AI. The decreasing contributions in emissivity of about $-20{\sim}40$ W m$^{-2}$ is caused by reductions in cloud cover and water vapor (Figs. 6f), which can be attributed to the common presence of high-pressure systems in subtropical arid areas (Zampieri et al. 2009) and less monsoon there. The decreasing contribution by lower atmospheric emissivity is compensated for by an increased contribution of about $+10{\sim}20$ W m$^{-2}$ by atmospheric heat storage that is caused by the generally warmer mean temperatures in arid regions.

### 4. Discussion and Conclusions

We found that the semiempirical equations of Brutsaert (1975) and Crawford and Duchon (1999) work very well to estimate the downwelling flux of longwave radiation by comparing these to estimates from observation, satellite, and reanalysis datasets, with $r^2$ ranging from 0.87 to 0.98 across the datasets for clear-sky and all-sky conditions. We then showed that one can use these equations to decompose this flux into different components, and relate changes to differences in cloud cover, water vapor, and lower atmospheric heat storage. We found that most diurnal changes in downwelling longwave radiation are caused by differences in lower atmospheric heat storage that are reflected in differences in surface air temperature, with the changes in atmospheric emissivity playing the secondary role. The dominance of surface air temperature can be also observed in the seasonal ranges of $R_{ld}$, except in tropical monsoon regions due to large variations in water vapor and cloud-cover. As for the spatial variation, from arid to humid region, the increasing lower atmospheric heat storage and decreasing atmospheric emissivity have an offsetting effect on the $R_{ld}$ variation, thus leading to relatively subtle changes in Rld along with aridity index.

Relating our decomposition to radiative kernel helps to gain a more comprehensive understanding of variations in $R_{ld}$. Referring to the sensitivity in the downwelling longwave radiation for an incremental change in an atmospheric property (e.g., $T_a$, $f_c$, and $e_a$), radiative kernel has been used to attribute $R_{ld}$ changes, based on numerically calculation with radiative transfer code (Previdi 2010 and Vargas Zeppetello et al. 2019) or partial differentiating with explicit formula for $R_{ld}$ (Shakespeare and Roderick, 2022). Following Shakespeare and Roderick (2022), the approximate radiative kernel of $T_a$, $f_c$, and $e_a$ are calculated based on Eqs. 8-9 ( i.e., $\frac{\partial R_{ld}}{\partial T} = 4\sigma\overline{\varepsilon}\,\overline{T_a}^{-3}$, $\frac{\partial R_{ld}}{\partial f_c} = \sigma\overline{T_a}^{-4} \times \left(1 - \overline{1.24\left(\frac{\overline{e_a}}{\overline{T_a}}\right)^{\frac{1}{7}}}\right)$, and $\frac{\partial R_{ld}}{\partial e_a} = \sigma\overline{T_a}^{-4} \times \frac{1.24}{7}\frac{(1-\overline{f_c})}{(\overline{e_a})^{\frac{6}{7}}(\overline{T_a})^{\frac{1}{7}}})$ and shown in the left panel of Fig. S8. As shown in Fig S8a, the sensitivity of $R_{ld}$ to $T_a$ peaks in the tropics with a maximum of around 5 W/m$^2$/K and decreases at higher latitudes, which is generally consistent with Shakespeare & Roderick (2022). Moreover, the seasonal cycle of the atmospheric properties themselves are shown in the right panel of Figure S8, which reveals that the spatial distribution of the contribution of

$T_a$, $e_a$, and $f_c$ to the seasonal variations in $R_{ld}$ (Figure 5) is dominated by the seasonal changes of the air
properties (Figs. S8b, S8d, and S8f) instead of the sensitivity of $R_{ld}$ to them (Figs. S8a, S8c, and S8e).
These equations can then be applied to different aspects of climate research. For instance, the values of
downwelling longwave radiation are often missing in FLUXNET data (Table S2), and these equations can
be used to fill the gaps with air temperature and humidity observations. We can also use these equations to
better understand the physical mechanisms for temperature change due to extreme events. For instance,
Park et al. (2015) and Alekseev et al. (2019) found that an enhancement of downwelling longwave radiation
in the Arctic is found to be preceded by the advection of moisture and heat. The equations by Brutsaert
(1975) and Crawford and Duchon (1999) can then be used to quantify the individual contributions by the
advection of heat and moisture (Tian et al. 2022). Another example is the attribution of differences in
temperature magnitudes across humid and arid regions (Ghausi et al., 2023). Du et al. (2020) used these
equations to explain why global warming was stronger during clear-sky conditions in observations in China
due to the greater sensitivity of clear-sky emissivity to a change in water vapor. This was then used to
explain the observed, stronger global warming in the arid regions of China, which have less clouds and a
higher frequency of clear-sky conditions than the humid regions. Furthermore, while the empirical
coefficient of 1.24 in Eq. (1) may change due to emissivity changes from greenhouse gases, this formulation
can nevertheless provide a useful basis in terms of the interannual changes of $R_{ld}$, which is shown in Fig.
S9. As shown in Fig. S9a, $R_{ld}$ increases in most of the land regions, at an average rate of 0.64 $W/m^2$/decade,
with the contribution of increased temperature, increased water vapor, and decreased cloud cover
contributing 0.46, 0.28, -0.10 $W/m^2$/decade, respectively. Furthermore, it can be observed in Figs. S9d-S9i
that the temperature effect is generally around 0.5 $W/m^2$/decade, while the influence of emissivity is
significantly dominant in the monsoon region, which is majorly due to the interannual changes in water
vapor.
It is worth noting that several effects on Rld variations are not included in B75 and C&D99, e.g., the well-
mixed greenhouse gas concentrations (Shakespeare and Roderick, 2022), large aerosol particles (Zhou and
Savijärvi. 2013), and cloud base (Viúdez-Mora et al. 2015). Although rarely influencing the diurnal change,
seasonal cycles, and spatial distribution, these terms needs attention when the interannual trend of Rld is
investigated under global warming, which can be implied by the difference between Figs. S9a and S9b. In
addition, B75 in conjunction with C&D99 is adopted in this work to decompose the Rld variations in
different spatial-temporal scales, considering its solid physical foundations and the relatively less
computation consumption. Further analysis can be performed based on other estimations, e.g. Prata 1996,
which shows consistency with reanalysis data (Allan et al. 2004). The cloud effect can be also detected
using the difference between all-sky and clear-sky Rld (Allan 2011; Ghausi et al., 2022). Moreover, datasets
that are more focused on radiation and energy budget can be used to test the robust of the results, e.g.,
BSRN (Driemel et al. 2018) and GEBA (Wild et al. 2017).
We conclude that the equations by Brutsaert (1975) and Crawford and Duchon (1999) are still very useful
in advancing our understanding of surface temperature changes. Our evaluation has shown how well these
equations estimate this flux, and our application to the decomposition of different contributions has shown
its utility in understanding the causes of its variation. These equations should help us to better understand
aspects of climate variability, extreme events, and global warming, linking these to the mechanistic
contributions by downwelling longwave radiation.

# Acknowledgments

This research is supported by the National Natural Science Foundation of China (52209026) and the Second
Tibetan Plateau Scientific Expedition and Research Program (grant no. 2019QZKK0208). This research
resulted from a research stay of YLT in AK's research group. This stay was supported by China Scholarship
Council as No. 202106210161. AK and SAG acknowledge funding from the Volkswagen Stiftung through
the ViTamins project.

# Author contributions

YLT, SAG, and AK conceived and designed the analysis, with inputs from DZ and GW. YLT performed
the analysis and discussed the results with all authors. YLT and AK wrote the paper.

# Competing interests

The contact author has declared that none of the authors has any competing interests.

# Data availability

The data used in this study was downloaded from the links provided with the references. No new data was
created.

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

**Figures**

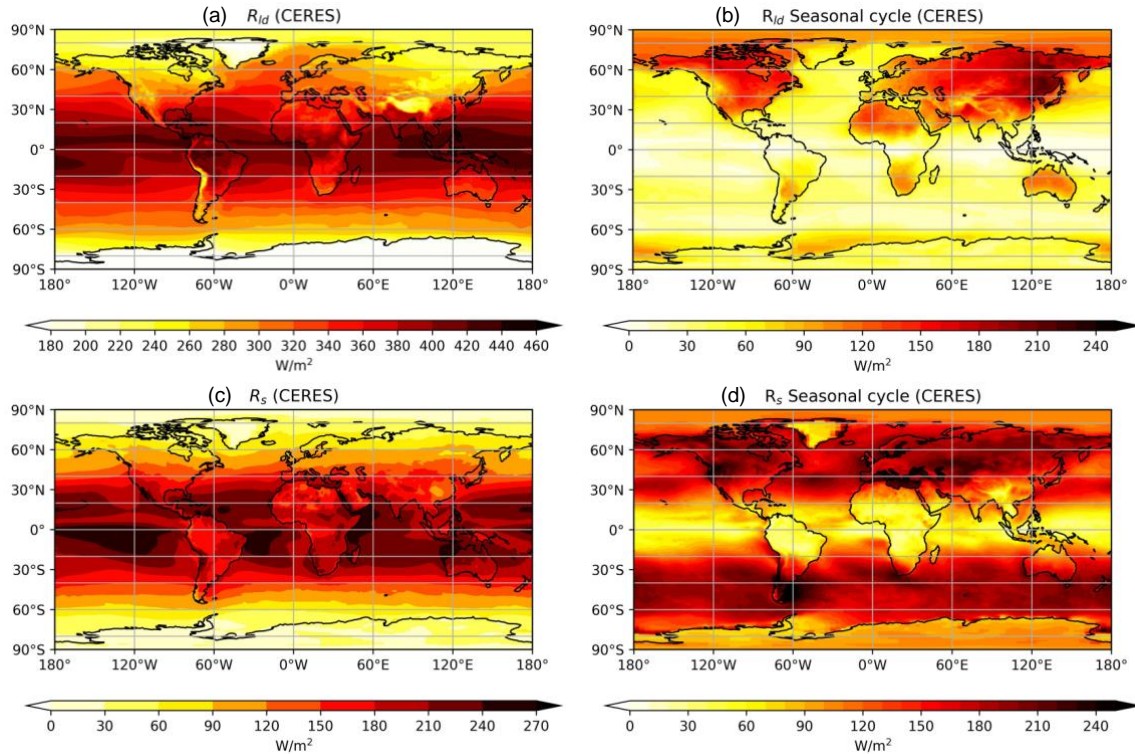

**Figure 1.** Spatial distribution of (a, c) the climatological mean and (b, d) the seasonal amplitude of
downward longwave radiation and absorbed solar radiation at the surface respectively from the NASA-
CERES dataset. The seasonal amplitude is calculated as the difference between the maximum and minimum
monthly data.

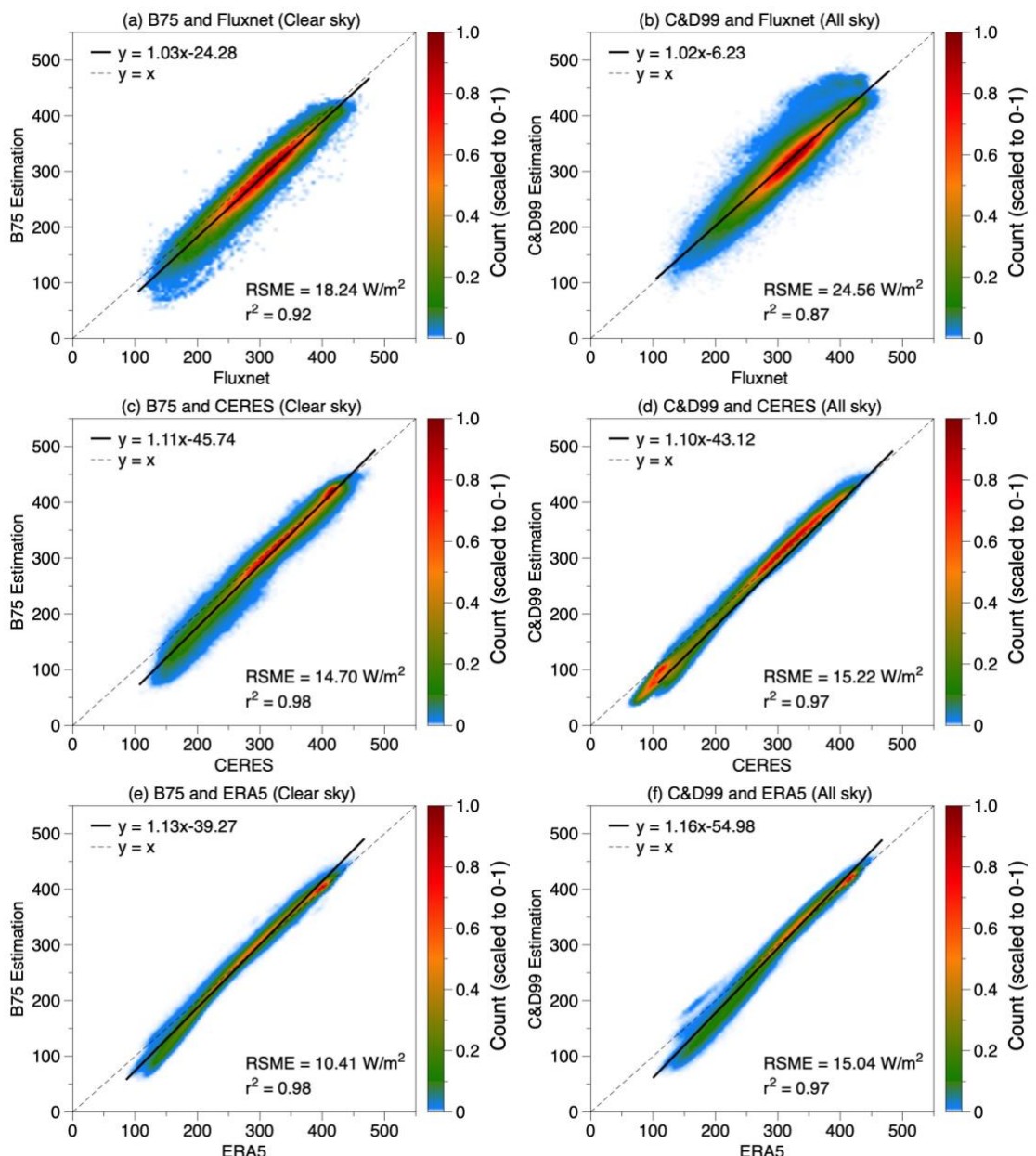


Figure 2. Comparison of Rld estimated by Brutsaert (1975) (a, c, e) for clear-sky conditions and by
Crawford and Duchon (1999) (b, d, f) for all-sky conditions using (a, b) FLUXNET hourly data of 189
sites, (c, d) NASA-CERES monthly data of 1°×1° from 2001 to 2018 and (e, f) ERA5 monthly data of
resolution of 1°×1° from 1979 to 2021. Colors indicate the density of the data points and is scaled to values
between 0 - 1.

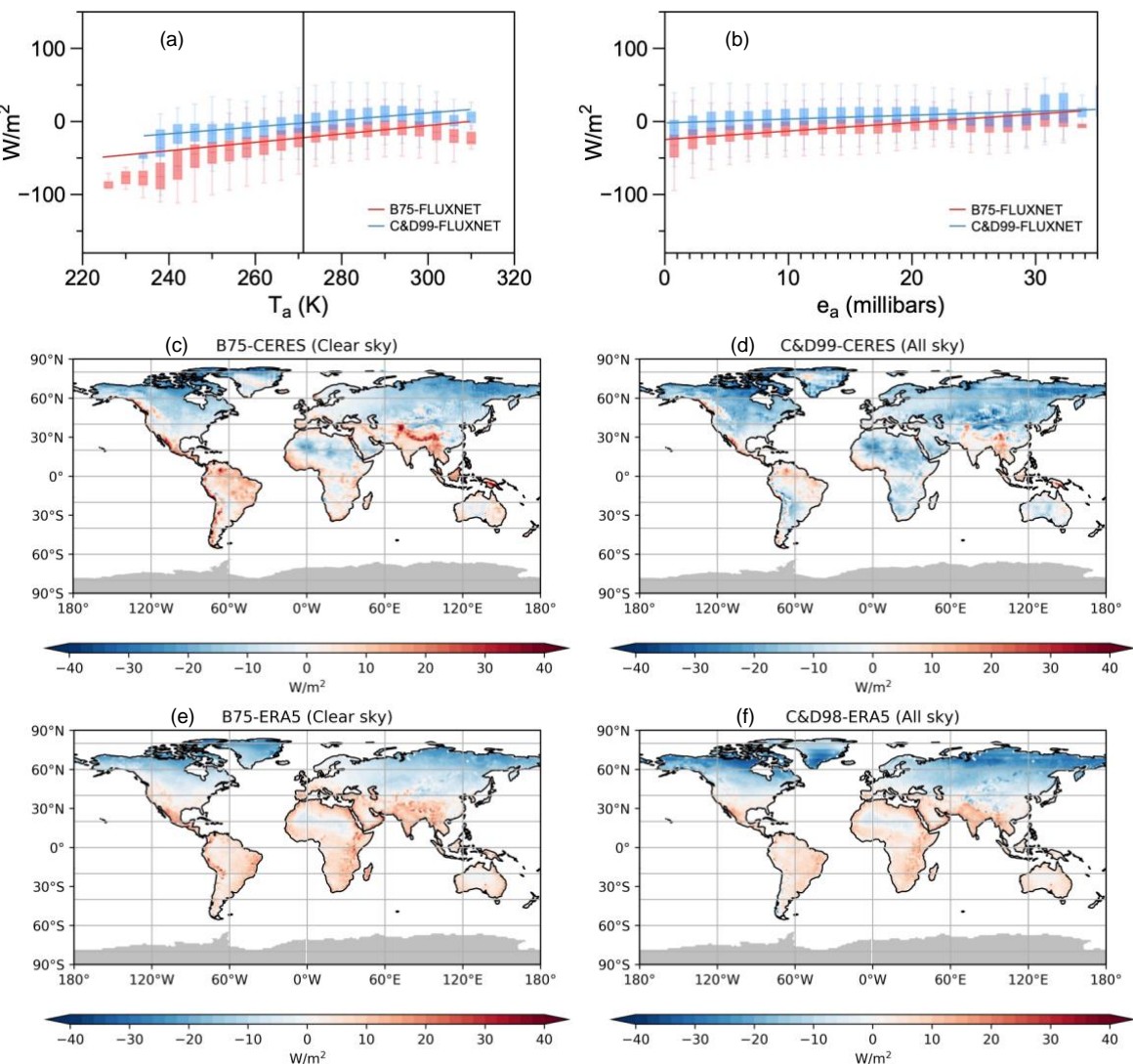

472

**Figure 3.** Biases in the estimates for multi-year mean $R_{ld}$ for FLUXNET data of 189 sites against (a) air temperature and (b) water vapor pressure. Distribution of biases in the estimates for multi-year mean $R_{ld}$ for (c, d) NASA-CERES data from 2001 to 2018 and (e, f) ERA reanalysis from 1979 to 2021 for (c, e) clear-sky and (d, f) all-sky conditions over land. Grey shading indicates missing values.


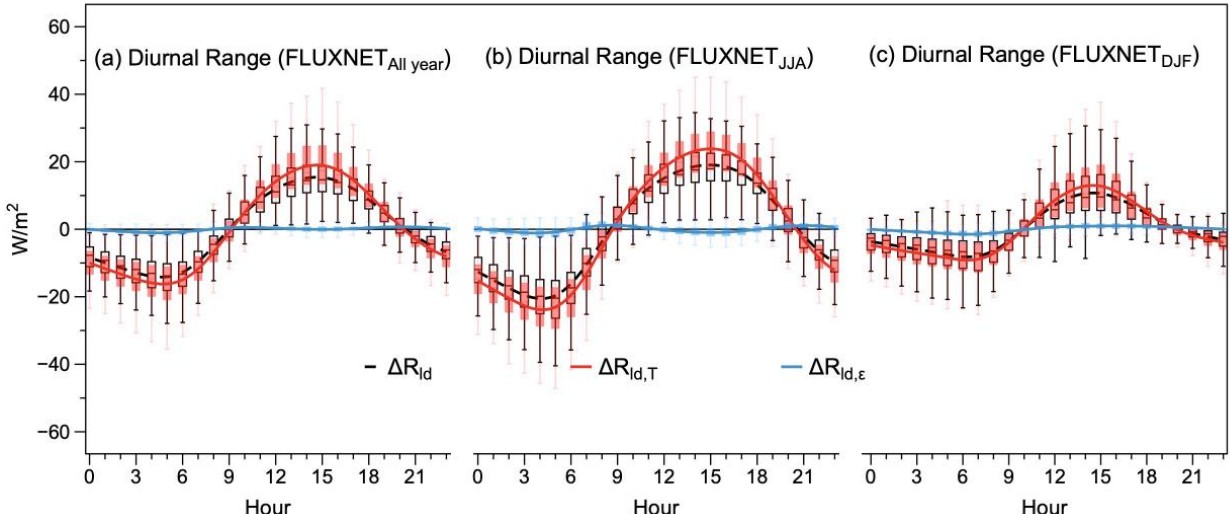

**Figure 4.** The multi-year average diurnal variations in $R_{ld}$ (black dashed line) and its decomposition into
contributions by changes in emissivity (blue, $\Delta R_{ld,\varepsilon}$) and lower atmospheric heat storage (red, $\Delta R_{ld,T}$) in
the FLUXNET dataset aggregated over 189 sites for (a) the whole year, (b) June-August, and (c) December
– February. The box shows the variation among the 189 sites. The upper and lower whiskers indicate 95[th]
and 5[th] percentiles, upper boundary, median line, and lower boundary of the box indicate the 75[th], 50[th], and
25[th] quantiles, respectively. For each site and each day, the daily mean value is removed, with the deviations
shown. Regression lines are based on site-mean or grid-mean value using LOESS regression.

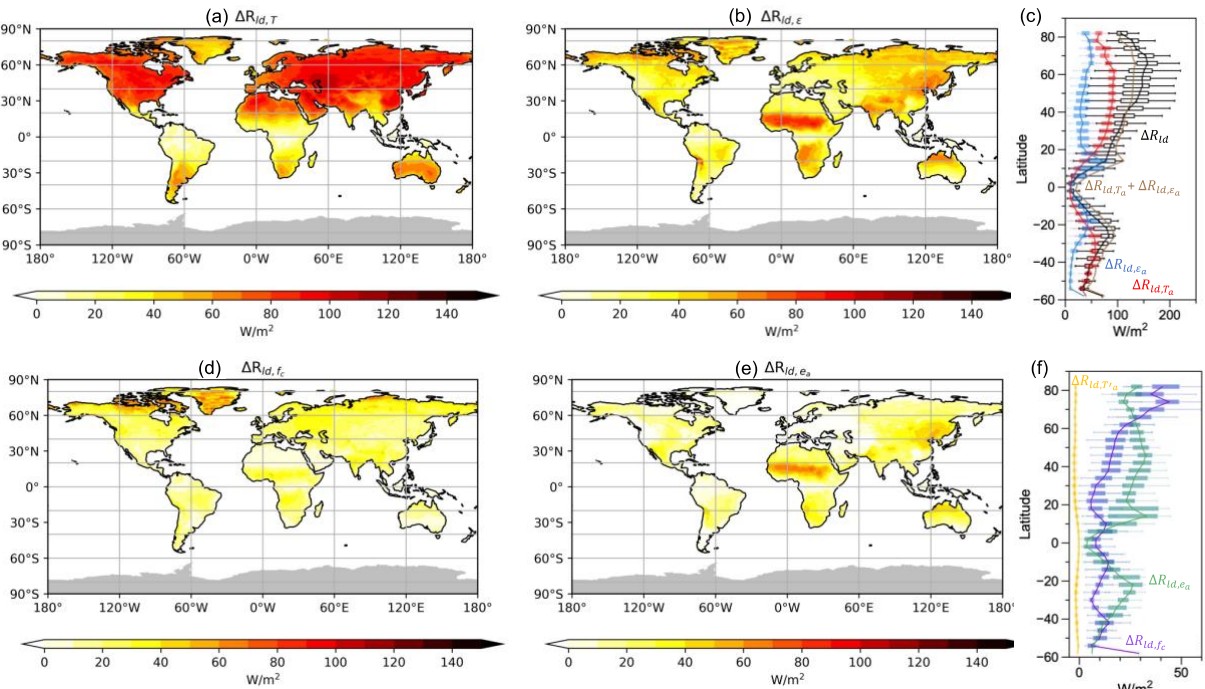

**Figure 5.** Decompositions of the mean seasonal variation (Δ, difference between the maximum and
minimum monthly data at each grid) of $R_{ld}$ in the NASA-CERES dataset into contributions by (a) lower
atmospheric heat storage ($\Delta R_{ld,T}$) and (b) emissivity ($\Delta R_{ld,\varepsilon}$), and (c) their latitudinal variations.
Decomposed of $\Delta R_{ld,\varepsilon}$ into contributions by variations in (d) cloud cover ($\Delta R_{ld,f_c}$) and (e) humidity

($\Delta R_{ld,e_a}$), (f) their latitudinal variations. In Figs. a, b, d, e, grey shading indicates missing values. In Figs. c and f, the box shows the variation among the land grids at the same latitude, while the solid line is their mean. The upper and lower whisker indicate 95[th] and 5[th] percentiles, upper boundary, median line, and lower boundary of the box indicate the 75[th], 50[th], 25[th] quantiles, respectively.

**Figure 6.** Decompositions of the multiyear-mean spatial variation of $R_{ld}$ (deviations of the multiyear-mean value for each grid from the land-mean value) in the NASA-CERES dataset into contributions by (a) lower atmospheric heat storage ($\Delta R_{ld,T}$) and (*b*) emissivity ($\Delta R_{ld,\varepsilon}$). Decomposition of $\Delta R_{ld,\varepsilon}$ into contributions by (c) variations in cloud cover ($\Delta R_{ld,f_c}$) and (d) humidity ($\Delta R_{ld,e_a}$). Ins Figs. a-d, grey shading indicates missing values. In Figs. e and f, the box shows the variation among the land grids with the same aridity. The upper and lower whisker indicate 95[th] and 5[th] percentiles, upper boundary, median line, and lower boundary of the box indicate the 75[th], 50[th], 25[th] quantiles, respectively.