# Peer review of "Understanding variations in downwelling longwave radiation using Brutsaert's equation"

_EGUsphere, 2023_

## Author Comment (AC1)

**A simple empirical model of surface downwelling longwave radiation (Rld) is evaluated against state of the art simulations and used to understand the driving factors influencing this important surface flux quantity.**

We thank the reviewer for the thoughtful assessment of our work and many insightful comments. We have now accordingly revised the manuscript with the point-by-point response mentioned in this document. The reviewer comments are shown in blue (bold) and our responses are shown in black. The text that is added in the revised manuscript is highlighted in Italic.

- **Major comment:**

**1) Others have applied or proposed empirical models of Rld which would be useful to refer to and discuss briefly e.g. Prata (1996) QJRMS doi:10.1002/qj.49712253306, Dilly & O'Brien (1998) QJRMS doi:10.1002/qj.49712454903 while there are also comparisons against reanalyses, more for clear-sky simulations e.g. Allan et al. (2004) JGR doi:10.1029/2004JD004816. Missing terms include well mixed greenhouse gas concentrations and large aerosol particles which can influence the Rld in drier atmospheres though may not contribute much for spatial/seasonal/diurnal variation discussed in the present study. Cloud base is also clearly important in determining the cloud radiative effect on Rld and the correlation between cloud and humidity could be implicitly included in the calibration of the equation. So some discussion of these issues along with limitations of the data sources would strengthen the manuscript in my opinion. There are a number of further specific points that can be considered.**

[Reply] Thank you for suggesting that referring to these studies on estimations and attribution of Rld variations will improve the manuscript. In addition, it is also necessary to make clear the limitations of our data sources. To accommodate this point, we have now revised the discussion section. Discussion is now added about the missing terms in B75 and C&D99 and the alternative methods/datasets that have been suggested.

[Action] We add a paragraph to clarify the limitation of the work in the Discussion section as follows, which also links to minor comments #7 and #11.

(1) We clarify the neglected terms in B75 and C&D99 as follows: *"It is worth noting that several effects on Rld variations are not included in B75 and C&D99, e.g., the well-mixed greenhouse gas concentrations (Shakespeare and Roderick, 2022), large aerosol particles (Zhou and Savijärvi. 2013), and cloud base (Viúdez-Mora et al. 2015). Although rarely influencing the diurnal change, seasonal cycles, and spatial distribution, these terms need attention when the interannual trend of Rld is investigated under global warming."*

(2) Combined with minor comment #7, we mention the alternative method to investigate the effect of Ta, ea, and cloud, which are the major drivers we focused on in this work: *"In this work, B75 in conjunction with C&D99 is adopted to decompose the Rld variations in different spatial-temporal scales, considering its solid physical foundations and the relatively less computation consumptions. Further analysis can be performed based on other estimations, e.g. Prata 1996,*

*which shows consistency with reanalysis data (Allan et al. 2004). In addition, the cloud effect can be also detected using the difference between all-sky and clear-sky Rld (Allan 2011).".*

(3) Combined with minor comment #11, we add a discussion about the data as follows: *"Moreover, datasets that are more focused on radiation and energy budget can be used to test the robustness of the results, e.g., BSRN (Driemel et al. 2018) and GEBA (Wild et al. 2017)."*

**2) Analysing the importance of the different drivers on changes in Rld from year to year may lead to differing conclusions to spatial/seasonal variation yet may be more relevant to long term climate change though were not assessed.**

[Figure]

Fig. R1 Decomposition of the interannual trend of $R_{ld}$ ($\Delta R_{ld}$) (a) to the interannual trend of surface air temperature ($\Delta R_{ld,T} = 4\bar{\varepsilon}\sigma\overline{T_a}^3\Delta T_a$) (d) and emissivity ($\Delta R_{ld,e_a} + \Delta R_{ld,f_c}$) (e), which composites of contributions of water vapor pressure ($\Delta R_{ld,e_a} = \sigma\overline{T_a}^4 \times \frac{1.24}{7}\frac{(1-\bar{f_c})}{(\overline{e_a})^{\frac{6}{7}}(\overline{T_a})^{\frac{1}{7}}})\Delta e_a$) (g) and cloud cover ($\Delta R_{ld,f_c} = \sigma\overline{T_a}^4 \times \left(1 - \overline{1.24\left(\frac{\overline{e_a}}{\overline{T_a}}\right)^{\frac{1}{7}}}\right)\Delta f_c$) (h), and their sum ($\Delta R_{ld,T} + \Delta R_{ld,e_a} + \Delta R_{ld,f_c}$) (b). Figs. c, f, and i show the corresponding latitudinal variations. $^{-}$ denotes the multi-year average, and $\Delta$ denotes the slope of linear regression of the yearly-mean data. In Figs. a, b, d, e, g, and h, grey shading indicates missing values. In Figs. c, f, and i, the box shows the variation among the land grids at the same latitude, while the solid line is their mean. The upper and lower whisker indicate 95th and 5th percentiles, upper boundary, median line, and lower boundary of the box indicate the 75th, 50th, 25th quantiles, respectively. Data are from the NASA-CERES dataset.

[Reply] Thank you for raising this point. To address it, we did additional analysis and applied the decomposition method to the interannual trend (obtained by linear regression) of global-land-mean Rld from 2000 to 2016 (See figure R1). As shown in Fig. R1a, $R_{ld}$ increases in most of the land regions, at an average rate of 0.64 W/m$^2$/decade, with the contribution of increased temperature, increased water vapor, and decreased cloud cover contributing 0.46, 0.28, -0.10 W/m$^2$/decade, respectively. The difference between Figs. R1a and R1b can be attributed to the changes in well-mixed greenhouse gas concentrations and large aerosol particles. Furthermore, it can be observed in Figs. R1d-R1i that temperature effect is generally around 0.5 W/m$^2$/decade, while the influence of emissivity is significantly dominant in the monsoon region, which is majorly due to the interannual trends in water vapor.

[Action] We add Fig. R1 to the supplementary material as Fig. S9 and discuss them in the Section Discussion.

- **Minor comment:**

**1) L20 - it should be stated if this is the spatial correlation which is much less stringent than for variability since there is a large amount of spatial autocorrelation yet large spatial ranges in driving terms**

[Reply] Thanks for raising this point. We agree that the characteristic of spatial correlation is worth noting. The correlation using the monthly data of CERES and ERA5 (Figs. R2c-R2f) might be influenced by the spatial correlation to some extent. However, the plot in Figs. R2a and R2b show the correlation based on hourly data of FLUXNET sites data, which are scattered locations and therefore have less geographical information but more temporal ones. As a result, the estimated high correlation in our study corresponds to agreement in both spatial and temporal variability.

[Action] To make this point clear, we have revised the sentence as "*We found a strong spatiotemporal correlation between estimated Rld and the datasets above, with r$^2$ ranging from 0.87 to 0.98 across the datasets for clear-sky and all-sky conditions.*" And Fig.R2 is Fig.2 in the revised text.

[Figure]

Figure R2. Comparison of Rld estimated by Brutsaert (1975) for clear-sky conditions (a, c, e) and by Crawford and Duchon (1999) for all-sky conditions (b, d, f) using FLUXNET hourly data of 189 sites (a, b), NASA-CERES monthly data of 1°×1° from 2001 to 2018 (c, d) and ERA5 monthly data of resolution of 1°×1° from 1979 to 2021 (e, f). Colors indicate the density of the data points and is scaled to values between 0 - 1.

**2) L23 - presumably cloud is correlated to humidity so is the contribution from cloud also implicitly including a contribution from humidity via the calibration of the empirical formula?**

[Reply] Thank you for the comment. First, we would like to make it clear that we have not calibrated the formula but used it as derived in B75 to understand the major causes of the $R_{ld}$

variations on the different temporal-spatial scales. In this approach, cloud and water vapor are treated as two major Greenhouse particles of humid air with different emissivity, which together with air temperature can be taken as the first order of the attribution of $R_{ld}$ variations. Under this situation, the influences of humidity on the cloud (and thus on $R_{ld}$) would be the second order of decomposition. To make the comparison of the contribution of different components fair, although cloud is no doubt closely correlated to atmospheric water vapor, we would like to separate the contributions of these two parts to focus on the first-order drivers of changes in $R_{ld}$.

**3) L24 - It should be stated that this refers to spatial variation in aridity from one region to another rather than changes over time. This is an important distinction to make since the conclusion may differ depending on whether changes are spatial or for the same location over time (either seasonally or interannual also may differ in character)**

[Reply and Action] Thanks for pointing this out. We have made the sentence clearer as the following "*We also found that as aridity increases over the region, the contributions from changes in emissivity and atmospheric heat storage tend to offset each other*".

**4) L33 - Although Rld is the dominant term, since it is strongly correlated with surface emitted longwave due to coupling of surface and near surface tempertaure, the net longwave flux is quite small e.g. Wild (2020) Clim Dyn, doi:10.1007/s00382-020-05282-7 and yet more strongly influenced by emissivity changes**

[Reply] Yes, the net longwave radiation is quite small. Here we specifically want to stress over the terms that contribute to radiative heating of the surface. We used the upward longwave radiation as a proxy of surface temperature, the remaining terms are then considered as energy input influencing surface temperature. This includes net solar radiation, downward longwave radiation, surface sensible and latent heat flux, and ground heat flux. Moreover, considering that most of the work is focused on the land region, we also provide the corresponding values over land according to Wild et al. 2015.

[Action] We revise the sentence as "*In the global mean surface energy budget, downward longwave radiation ($R_{ld}$) is dominant surface energy input (333 W/m$^2$ in global mean and 306 W$^2$/m over land), contributing around twice as much energy as absorbed solar radiation (161 W/m$^2$ in global mean and 184 W$^2$/m over land) (Trenberth et al. 2009, Wild et al. 2015).*".

**5) eq(4) - when cloud fraction is 100%, an emissivity of 1 is implied yet high altitude cloud will not emit much longwave to the surface. The authors should comment on the deficiencies in the empirical formulation**

[Reply] Thank you for raising this point. We acknowledge the importance of cloud height in determining the downward longwave radiation, particularly for the optically thick clouds (Viúdez-Mora et al. 2014)., which is one of the biggest deficiencies of B75 and C&D99.

Apart from commenting on it in discussion, we also to some extent address this issue by calculating the cloud cover data using the specific way provided in C&D99 (fc' = 1-R/Rpot, R and Rpot are the solar radiation and the potential solar radiation respectively), which can directly reflect the influence of cloud on radiations. We plot fc' against the cloud data from CERES (fc) as shown in Fig. R3. We find that fc' is generally smaller than fc, with an average difference of 0.3, and the maximum of fc' is around 0.6 instead of 1. Furthermore, when estimating Rld using fc' from Equation 3 under all sky conditions, the error distribution closely resembles that of clear sky conditions (Figure R4d and R4f), albeit with generally larger values. Moreover, this discrepancy affects the decomposition of seasonal and spatial variations, highlighting a higher contribution of water vapor anomalies to emissivity in tropical regions, particularly the Intertropical Convergence Zone, while cloud cover variations exert a greater influence over high latitudes (Figs. R5 and R7).

[Action]

(1) We stress in the discussion the necessity to consider the cloud height because high-altitude clouds will emit relatively less Rld compared to low-altitude ones.

(2) We have repeated all our analyses using fc calculated from Equation 3 instead of the cloud cover fraction. The related part in the method is revised as follows: "*Cloud cover $f_c$ is calculated using Eq. (3) for all three datasets with incoming solar radiation at the surface ($R_s$) and the potential solar radiation ($R_{s,pot}$).*" Fig.R4 is Fig.3 in the revised text.

[Figure]

Fig. R3 The scatter plot using the calculated cloud cover fraction (fc'=1-R/Rpot) and cloud cover from CERES datasets (fc). The dashed orange line is the linear regression while the solid blue line delicates the slope of 1.

[Figure]

Fig. R4 Biases in the estimates for multi-year mean $R_{ld}$ for FLUXNET data of 189 sites against air temperature (a) and water vapor pressure (b). Distribution of biases in the estimates for multi-year mean $R_{ld}$ for (c, d) NASA-CERES data from 2001 to 2018 and (e, f) ERA reanalysis from 1979 to 2021 for (c, e) clear-sky and (d, f) all-sky conditions over land. Grey shading indicates missing values.

**6) L62 - what is deemed "very good" agreement?**

[Reply and action] Thank you for the comment. We have now made the sentence clearer by adding the quantification as "*with the r² of 0.883 and RMSE of 15.367 W/m²*".

**7) L67 Effects of cloud on surface longwave flux can also be estimated from cloudy minus clear-sky flux calculations e.g. Allan (2011) Met Apps doi:10.1002/met.285**

[Reply] Thanks for suggesting a good point. We have added it in the discussion.

[Action] We add related text in the Section Discussion as in the reply to major comment #1.

**8) L68 - in humid regions, since much of the longwave spectrum is saturated, much of the downward emission will be determined by the near surface temperature which may or may not reflect the heat content of the atmosphere and can also differ from the surface skin temperarture quite markedly leading to a large sensitivity to how this emitting temperature is defined e.g. Raisenen (1996) Tellus doi:10.1034/j.1600-0870.1996.t01-2-00004.x; Allan (2000) J. Clim doi:10.1175/1520-0442(2000)013<1951:EOSCSL>2.0.CO;2**

[Reply] We agree that in humid regions the downward long-wave radiation is strongly determined by near-surface air temperature. This has also been shown by other studies (For eg., Vargas Zeppetello et al., 2019). However, the near-surface air temperature is in turn shaped by changes in lower atmospheric heat storage (Panwar et al., 2019; Panwar et al., 2022). As a result, the changes in decomposed Rld associated with air temperature mostly represent changes in lower atmospheric heat storage. To make it clear in the main text, we have changed the term "heat content of the atmosphere "to "lower-atmospheric heat storage".

[Action] The sentence is revised as *"This expression for downwelling longwave radiation Rld given by Eqn. (5) allows us to quantify the different contributions by cloud cover, fc, water vapor concentrations, ea (as a measure of the total water vapor content of the lower layer of atmospheric column), and air temperature, Ta (as a proxy for the heat storage within the lower atmosphere)"*. Also, we have stressed the "lower" ahead of "atmospheric heat storage" in the whole manuscript.

**9) L83 - since near surface temperature is more physically related to Rld than atmospheric heat storage, the referal to heat storage is not needed and not useful in my opinion.**

[Reply and action] As mentioned in the reply to comment 8, all the related expressions are revised as "the lower atmosphere".

**10) L80 - full attribution would involve radiative transfer calculations which are more accurate than an empirical formula**

[Reply] Thank the reviewer for the constructive comment. Yes radiative transfer calculation will be more accurate but to accommodate this suggestion in part, we have added comparison with radiative kernels. Defined as $\frac{\partial R_{ld}}{\partial T}$ and $\frac{\partial R_{ld}}{\partial q}$, radiative kernels for $T$ and $q$ are respectively typically calculated numerically using a radiative transfer code and high-resolution 3D climate model output (e.g., Previdi 2010 and Vargas Zeppetello et al. 2019). Apart from the climate model, the radiative kernels can be equivalently obtained with an explicit formula for $R_{ld}$ and a partial differentiating method, as demonstrated in Shakespeare & Roderick (2022). Following Shakespeare & Roderick (2022), we calculated the radiative kernels of $T_a$, $f_c$, and $e_a$ based on

B75 and C&D99 in terms of seasonal change, i.e., $\frac{\partial R_{ld}}{\partial T} = 4\sigma\overline{\epsilon}\overline{T}_a^{\ 3}, \frac{\partial R_{ld}}{\partial f_c} = \sigma\overline{T}_a^{\ 4} \times \left( 1 - \right.$

$$\overline{1.24\left(\frac{\overline{e_a}}{\overline{T_a}}\right)^{\frac{1}{7}}}, \text{ and } \frac{\partial R_{ld}}{\partial e_a} = \sigma \overline{T_a}^4 \times \frac{1.24}{7}\frac{(1-\overline{f_c})}{(\overline{e_a})^{\frac{6}{7}}(\overline{T_a})^{\frac{1}{7}}},$$ which are shown in Figure R5, together with the

seasonal cycle of the atmospheric properties themselves. For a clearer interpretation, we also plot the contribution of $T_a$, $f_c$, and $e_a$ in this reply as Fig. R6.

As shown in Fig R5a, the sensitivity of $R_{ld}$ to $T_a$ peaks in the tropics with a maximum of around 5 W/m$^2$/K and decreases at higher latitudes, which is generally consistent with Shakespeare & Roderick (2022). Moreover, the seasonal cycle of the atmospheric properties themselves are shown in the right panel of Figure R5, which reveals that the spatial distribution of the contribution of $T_a$, $e_a$, and $f_c$ to the seasonal variations in $R_{ld}$ (Figure R6) is dominated by the seasonal changes of the air properties (Figs. R5b, R5d, and R5f) instead of the sensitivity of $R_{ld}$ to them (Figs. R5a, R5c, and R5e).

[Action] We add Fig. R5 to the supplementary information as Fig.S8, put Fig.R6 in the main text as Fig. 5, and add the related analysis to the discussion.

[Figure]

Fig. R5 Distribution of the sensitivity of the seasonal cycle of $R_{ld}$ to surface air temperature ($\frac{\partial R_{ld}}{\partial T} = 4\sigma\overline{\varepsilon}\overline{T_a}^3$) (a),

cloud cover ($\frac{\partial R_{ld}}{\partial f_c} = \sigma \overline{T_a}^4 \times \left(1 - \overline{1.24\left(\frac{\overline{e_a}}{\overline{T_a}}\right)^{\frac{1}{7}}}\right)$) (c), and water vapor pressure ($\frac{\partial R_{ld}}{\partial e_a} = \sigma \overline{T_a}^4 \times \frac{1.24}{7}\frac{(1-\overline{f_c})}{(\overline{e_a})^{\frac{6}{7}}(\overline{T_a})^{\frac{1}{7}}}$) (e), and

their latitudinal variations. Distribution of the seasonal cycle of surface air temperature (b), cloud cover (d), and water vapor pressure (f), and their latitudinal variations. Seasonal cycle (Δ) indicates the difference between the

maximum and minimum montly data. In maps, grey shading indicate missing values. In boxplots, the box shows the variation among the land grids at the same latitude, while the solid line is their mean. The upper and lower whisker indicate 95[th] and 5[th] percentiles, upper boundary, median line, and lower boundary of the box indicate the 75[th], 50[th], 25[th] quantiles, respectively. Data are from the NASA-CERES dataset.

[Figure]

Fig. R6 Decompositions of the mean seasonal variation of $R_{ld}$ ($\Delta$, difference between the maximum and minimum monthly data at each grid) in the NASA-CERES dataset into contributions by lower-level atmospheric heat storage ($\Delta R_{ld,T}$) (a) and emissivity ($\Delta R_{ld,\varepsilon}$) (b), and their latitudinal variations (c). Decomposed of $\Delta R_{ld,\varepsilon}$ into contributions by variations in cloud cover ($\Delta R_{ld,f_c}$) (d) and humidity ($\Delta R_{ld,e_a}$) (e), their latitudinal variations (f). In Figs. a, b, d, e, grey shading indicates missing values. In Figs. c and f, the box shows the variation among the land grids at the same latitude, while the solid line is their mean. The upper and lower whisker indicate 95[th] and 5[th] percentiles, upper boundary, median line, and lower boundary of the box indicate the 75[th], 50[th], 25[th] quantiles, respectively.

**11) Section 2 can be improved by stating limitations and accuracies of the data and whether the analysis is restricted to monthly or hourly data. How is missing FLUXNET data accounted for, particularly in relation to sampling of the diurnal and seasonal cycle? Did the authors consider BSRN data which provide well-calibrated Rld estimates from a number of sites or GEBA which provides more sites with a lower level of quality control e.g. Wild et al. (2017) ESSD doi:10.5194/essd-9-601-2017 or are these included in FLUXNET?**

[Reply] Thanks for the suggestion. The validation of the formula is based on both half-hourly data (FLUXNET) and monthly data (ERA5 and CERES data). The decomposition of the diurnal range, seasonal range, spatial variation, and interannual trends are based on half-hourly data (FLUXNET), monthly data (CERES), multi-year mean data (CERES), and yearly data (CERES). In FLUXNET data, the gap filling is done using the multidimensional scaling method (MDS, Reichstein et al. 2005). However, we agree that further investigation based on the datasets that are more focused on energy and radiation will help to test the robustness of the results, e.g. BSRN/GEBA.

[Action] We make these two points clearer and more understandable in Section 2. Also, as we reply to the major comment #1, we mention in Section 2 that further investigation can be performed based on BSRN/GEBA datasets.

**12) Figure 2 - is this monthly gridpoint data, or climatological mean or also higher resolution daily/hourly data. This information along with the time period considered seem necessary**

[Reply] The FLUXNET data is half-hourly data, while the ERA5 and CERES data are monthly data.

[Action] We revise the caption to make it clearer as in Fig. R4.

**13) Figure 4 - is this multiannual average and if so for what years? It would also be useful to show the actual dTa and dε. Were spatial maps of the maximum minus minimum considered like in Figure 5?**

[Figure]

Fig. R7 The multi-year average diurnal variations in $R_{ld}$ (black dashed line) and its decomposition into contributions by changes in emissivity (blue, $\Delta R_{ld,\varepsilon}$) and lower-level atmospheric heat storage (red, $\Delta R_{ld,T}$) in the FLUXNET dataset aggregated over 189 sites for the whole year (a), June-August (b), and December – February (c). The box shows the variation among the 189 sites. The upper and lower whisker indicate 95th and 5th percentiles, upper boundary, median line, and lower boundary of the box indicate the 75th, 50th, 25th quantiles, respectively. For each site and each day, the daily mean value is removed, with the deviations shown. Regression lines are based on site-mean or grid-mean value using LOESS regression.

[Figure]

[Figure]

Fig. R8 The multi-year mean diurnal variations in $T_a$ (a) and water vapor pressure (b) in the FLUXNET dataset aggregated over 189 sites. The box shows the variation among the 189 FLUXNET sites. The upper and lower whisker indicate 95th and 5th percentiles, upper boundary, median line, and lower boundary of the box indicate the 75th, 50th, 25th quantiles, respectively. The solid lines are Loess fit.

[Reply] We apologize for not making it clear. Yes, Figure R7 (Figure 4 in the previous version of the manuscript) is the multiannual average. The time duration depends on the data availability of each FLUXNET site. Following the comment, we plot the actual diurnal variations of air temperature and emissivity in Fig. R8. Since Figs. R7-R8 are based on FLUXNET site data, which are generally scattered in Europe and North United States (as shown in the reply to the next comment), we think it is more interpretable to show the results in a boxplot as Figs. R7-R8 instead of maps.

[Action] To make the time scale of the data clearer, we revised the caption as the one of Fig. R7, which is Fig.4 in the revised text. We add Fig. R8 to the supplementary material as Fig. S4 and add the related analysis to the decomposition of the diurnal range of Rld.

**14) L158 - a map of FLUXNET coverage at the beginning would be useful (perhaps as dots on Figure 1 or an extra panel)**

[Figure]

Fig. R9 Locations of 189 FLUXNET sites

[Reply and actions] It is a useful comment! Following this, we add the map of the FLUXNET site locations (Fig. R9) into the supplementary material as Fig. S2.

**15) Figure 5 - I am used to seeing blue/red color scale to denote positive and negative values so a single color scale may be more appropriate. A total dRld map would also be useful and perhaps a residual in case the terms do not add up to the total or alternatively a map showing the dominant term in each region (T, fc, εcs)**

[Reply] Yes, we agree that diverging colors may give a wrong interpretation of the figure. Also, the comparison between the actual dRld and a sum of the contribution dT, dfc, dea will be helpful.

[Action] We have changed the diverging color bars to sequential color bars as in Figs R6. We also add $dR_{ld,T}+dR_{ld,fc}+dR_{ld,ea}$ to Fig. R6c (the brown line) to compare with the actual $dR_{ld}$ (the solid black line), which shows the changes of T, fc, ea can together explain more than 95% of the seasonal range of Rld.

**16) L164 - it is not surprising that areas with very large seasonal temperature changes produce large changes in Rld and the changes are themselves determined by the very large downward solar changes. Over monsoon regions I expect that there is some compensation as it moves from hot/dry/clear to cool/moist/cloudy. Again, the mean dTa and dε would be useful to show. Cloud and humidity are correlated so I wonder if this effect accentuates the apparent influence of cloud?**

[Reply] Thank you for this constructive comment. It is true that over the monsoon regions, the contribution of the emissivity changes to the Rld seasonal cycle is much more dominant, which is majorly due to the large seasonal variation of the water vapor (Fig. R6). As we reply to minor comment #2, these results are clearer because we now calculate the cloud cover according to the definition of C&D99 (fc') instead of using cloud data of CERES (fc), because fc' is better at reflecting the impact of clouds on radiation. Furthermore, we agree that a mean dTa and dε will be useful and show them in the right panel of Fig. R5, which to some extent suggests that the seasonal variation of cloud and water vapor is not necessarily coupled. Also, considering the different emissivity, we think these two contributions should be considered separately.

[Action] We revise the analysis of the decomposition of the seasonal cycle according to Fig. R6 and add the map of the mean seasonal range of dTa and dε (Fig. R5) to the supplementary.

**17) L174 - I did not completely understand this sentence**

[Reply] We apologize for not making it clear. This sentence is meant to explain the general less seasonal variation of Rld over marine than over land. Over the land, the changes in radiation are majorly buffered by the heat storage in the lower atmosphere by the variations in convective boundary layer height. However, over marine areas, solar radiation penetrates the transparent water bodies, the heat storage of which hence buffers the season cycle of the radiation over the ocean. Since the heat storage of the water body is larger than that of the lower atmospheric boundary layer, the buffering effect is consequently larger, which leads to the less seasonal cycle of the surface temperature and Rld over the ocean.

[Action] We revise the sentence as what we explain in the reply.

**18) Figure 6 - the caption needs more explanation. Are the map values the dRld compared to global (land) mean? I did not find the lower plots compelling since it is obvious the temperature effect relates to latitude (strength of the sun) and has no simple bearing on aridity so inferring relationships between temperature effect on aridity seem misleading (error bars are very large compared to the variation across AI). Contours of Aridity Index may be useful for interpretation**

[Reply] Yes, the map values are deviations of the multiyear-mean value for each grid from the land-mean value. We agree the map of aridity will contribute to the analysis. Although the global distribution of temperature is closely related to latitude, the influence of aridity on temperature variation has been explicitly revealed and thermodynamically explained in Ghausi et al. 2023. Therefore, we think it is worthwhile to analyze the spatial variation of temperature effect against aridity.

[Action] We clarify the map values clearly in the corresponding caption as shown in Fig. R8. Moreover, we also provide the variations in Ta, ea, and fc along with the aridity (Fig. R9) and the spatial distribution of the aridity (Fig. R10) in the supplementary for a more straightforward interpretation.

[Figure]

Fig. R8 Decompositions of the multiyear-mean spatial variation (Δ, deviations of the multiyear-mean value for each grid from the land-mean value) of $R_{ld}$ in the NASA-CERES dataset into contributions by lower-level atmospheric heat

storage ($\Delta R_{ld,T}$) (a) and emissivity ($\Delta R_{ld,\varepsilon}$) ($b$). Decomposition of $\Delta R_{ld,\varepsilon}$ into contributions by variations in cloud cover ($\Delta R_{ld,f_c}$) (c) and humidity ($\Delta R_{ld,e_a}$) (d). Ins Figs a-d, grey shading indicates missing values. In Fig. e and f, the box shows the variation among the land grids with the same aridity. The upper and lower whisker indicate 95th and 5th percentiles, upper boundary, median line, and lower boundary of the box indicate the 75th, 50th, 25th quantiles, respectively. Lines are linear regression.

[Figure]

Fig. R9 The variations along the aridity of the multiyear-mean surface air temperature (a), surface water vapor pressure (b), and cloud cover (c). Δ means deviations of the multiyear-mean value for each grid from the land-mean value. The upper and lower whisker indicate 95th and 5th percentiles, upper boundary, median line, and lower boundary of the box indicate the 75th, 50th, 25th quantiles, respectively. Lines are linear regression. Data is from NASA-CERES.

[Figure]

Fig. R10 Distribution of the aridity over land. Data is from NASA-CERES.

**19) L209/L230 - "very well" and "very useful" are quite qualitative descriptions - how good is good enough? How does it compare to observational accuracy?**

[Reply] Thanks for the comment. We have now tried to make it clear that our interpretation is not to say that B75 and C&D99 are better than the observation or other model products in terms of accuracy but rather focus on their ability to reveal the spatial-temporal variation with a more solid physical background.

[Action] Accordingly, L209 is revised as "*We found that the semiempirical equations of Brutsaert (1975) and Crawford and Duchon (1999) work very well to estimate the downwelling flux of longwave radiation by comparing these to estimates from observation, satellite, and reanalysis datasets, with the $R^2$ higher than 0.85*". L230 is revised as "*We conclude that the equations by*

*Brutsaert (1975) and Crawford and Duchon (1999) are still very useful to advance our understanding of the diurnal, seasonal, and multiyear-mean spatial variation in Rld.".*

**20) Summary - a missing component of the work is to look at interannual changes over time and how tempertaure and humidity determine year to year changes in Rld. This may be quite different to seasonal and spatial changes which are strongly determined by solar forcing and yet be more relevant to longer term climate change.**

[Reply and action] Thank you for this suggestion. As we reply to the major comment #1, we have conducted the analysis of interannual changes and add the related figures and text to the manuscript.

[References]

Allan, R. P., Ringer, M. A., Pamment, J. A., and Slingo, A. (2004), Simulation of the Earth's radiation budget by the European Centre for Medium-Range Weather Forecasts 40-year reanalysis (ERA40), J. Geophys. Res., 109, D18107, https://doi.org/10.1029/2004JD004816 .

Driemel, A., Augustine, J., Behrens, K., Colle, S., et al. (2018) Baseline Surface Radiation Network (BSRN): structure and data description (1992–2017), Earth Syst. Sci. Data, 10, 1491–1501, https://doi.org/10.5194/essd-10-1491-2018 .

Ghausi, S. A., Tian Y., Zehe E., & Kleidon A. (2023) Radiative controls by clouds and thermodynamics shape surface temperatures and turbulent fluxes over land. Proceedings of the National Academy of Sciences. 120 (29), e2220400120. https://doi.org/10.1073/pnas.2220400120

Prata, A.J. (1996), A new long-wave formula for estimating downward clear-sky radiation at the surface. Q.J.R. Meteorol. Soc., 122: 1127-1151. https://doi.org/10.1002/qj.49712253306

Panwar, A., Kleidon, A., & Renner, M. (2019). Do surface and air temperatures contain similar imprints of evaporative conditions? Geophysical Research Letters, 46, 3802–3809. https://doi.org/10.1029/2019GL082248

Panwar, A., and A. Kleidon, 2022: Evaluating the Response of Diurnal Variations in Surface and Air Temperature to Evaporative Conditions across Vegetation Types in FLUXNET and ERA5. J. Climate, 35, 6301–6328, https://doi.org/10.1175/JCLI-D-21-0345.1.

Previdi, M. (2010). Radiative feedbacks on global precipitation. Enviromental Research Letters, 5, 025211. https://doi.org/10.1088/1748-9326/5/2/025211

Reichstein, M., Falge, E., Baldocchi, et al. (2005), On the separation of net ecosystem exchange into assimilation and ecosystem respiration: review and improved algorithm. Global Change Biology, 11: 1424-1439. https://doi.org/10.1111/j.1365-2486.2005.001002.x

Shakespeare C. J. and M. Roderick. (2022). Diagnosing Instantaneous Forcing and Feedbacks of Downwelling Longwave Radiation at the Surface: A Simple Methodology and Its Application to CMIP5 Models. Journal of Climate.

Vargas Zeppetello, L. R., Donohoe, A., & Battisti, D. S. (2019). Does surface temperature respond to or determine downwelling longwave radiation? Geophysical Research Letters, 46, 2781–2789. https://doi.org/10.1029/2019GL082220

Viúdez-Mora, A., Costa-Surós, M., Calbó, J., and González, J. A. (2015), Modeling atmospheric longwave radiation at the surface during overcast skies: The role of cloud base height, J. Geophys. Res. Atmos., 120, 199–214, https://doi.org/10.1002/2014JD022310

Wild, M., Folini, D., Hakuba, M.Z. et al. The energy balance over land and oceans: an assessment based on direct observations and CMIP5 climate models. Clim Dyn 44, 3393–3429 (2015). https://doi.org/10.1007/s00382-014-2430-z

Wild, M., Ohmura, A., Schär, C., Müller, G., Folini, D., Schwarz, M., Hakuba, M. Z., and Sanchez-Lorenzo, A.: The Global Energy Balance Archive (GEBA) version 2017: a database for worldwide measured surface energy fluxes, Earth Syst. Sci. Data, 9, 601–613, https://doi.org/10.5194/essd-9-601-2017, 2017.

Zhou and Savijärvi. 2014. The effect of aerosols on long wave radiation and global warming. Atmospheric Research, 135–136: 102-111 https://doi.org/10.1016/j.atmosres.2013.08.009

---

## Author Comment (AC2)

**This is a clear manuscript that cleanly demonstrates the utility of Brutsaert's equation for calculating DLR from two easily observable state variables (two meter air temperature and vapor pressure). I had never heard of Brutsaert's empirical equation before reading this paper, and the manuscript gave some nice physical insights on DLR variations across time and space.**

We thank the reviewer for his appreciation of our work and constructive comments. The reviewer suggested some important points that need more clarity. We have now accordingly revised the manuscript with the point-by-point response mentioned in this document. The reviewer comments are shown in bold blue and our responses are shown in black. And the italic text in Times New Roman style is what we revised in the text.

**One major question I have is, how does Brutsaert's equation compare to radiative kernels, which have been used by a few recent studies (e.g. Shakespeare & Roderick 2022 and Vargas Zeppetello et al. 2019) to attribute changes in DLR to changes in near-surface state variables? This might help give Brustaert's equation a more interpretable physical intuition, because the kernels are calculated via climate models and their radiative transfer schemes.**

[Reply] Thank the reviewer for the constructive comment. As suggested, we have now used radiative kernels as described in Vargas Zeppetello et al. (2019) and Shakespeare & Roderick (2022) to understand the sensitivity in the estimated downwelling longwave radiation for the seasonal change in the atmospheric property ($T_a$, $e_a$, and $f_c$), i.e., $\frac{\partial R_{ld}}{\partial T} = 4\sigma\bar{\varepsilon}\bar{T}_a^3$, $\frac{\partial R_{ld}}{\partial f_c} =$

$\sigma\bar{T}_a^4 \times \left(1 - \overline{1.24\left(\frac{\overline{e_a}}{\overline{T_a}}\right)^{\frac{1}{7}}}\right)$, and $\frac{\partial R_{ld}}{\partial e_a} = \sigma\bar{T}_a^4 \times \frac{1.24}{7}\frac{(1-\bar{f_c})}{(\overline{e_a})^{\frac{6}{7}}(\overline{T_a})^{\frac{1}{7}}}$. The results are shown in Figure R1, together with the seasonal cycle of the atmospheric properties themselves. For a clearer interpretation, we also plot the contribution of $T_a$, $f_c$, and $e_a$ to the seasonal change of Rld in this reply as Fig. R2.

As shown in Fig R1a, the sensitivity of $R_{ld}$ to $T_a$ peaks in the tropics with a maximum of around 5 W/m$^2$/K and decreases at higher latitudes, which is generally consistent with Shakespeare & Roderick (2022). Moreover, the seasonal cycle of the atmospheric properties themselves is shown in the right panel of Figure R1, which reveals that the spatial distribution of the contribution of $T_a$, $e_a$, and $f_c$ to the seasonal variations in $R_{ld}$ (Figure R2) is dominated by the seasonal changes of the air properties (Figs. R1b, R1d, and R1f) instead of the sensitivity of $R_{ld}$ to them (Figs. R1a, R1c, and R1e)."

[Action] We add Fig. R1 to the supplementary information as Fig.S8, put Fig.R2 in the main text as Fig. 5, and add the related analysis to the discussion:

[Figure]

Fig. R1 Distribution of the sensitivity of the seasonal cycle of $R_{ld}$ to surface air temperature ($\frac{\partial R_{ld}}{\partial T} = 4\sigma\bar{\varepsilon}\overline{T_a}^3$) (a),

cloud cover ($\frac{\partial R_{ld}}{\partial f_c} = \sigma\overline{T_a}^4 \times \left(1 - \overline{1.24\left(\frac{\overline{e_a}}{\overline{T_a}}\right)^{\frac{1}{7}}}\right)$) (c), and water vapor pressure ($\frac{\partial R_{ld}}{\partial e_a} = \sigma\overline{T_a}^4 \times \frac{1.24}{7}\frac{(1-\bar{f_c})}{(\overline{e_a})^{\frac{6}{7}}(\overline{T_a})^{\frac{1}{7}}}$) ($e$), and

their latitudinal variations. Distribution of the seasonal cycle of surface air temperature (b), cloud cover (d), and water vapor pressure (f), and their latitudinal variations. Seasonal cycle ($\Delta$) indicates the difference between the maximum and minimum montly data. In maps, grey shading indicate missing values. In boxplots, the box shows the variation among the land grids at the same latitude, while the solid line is their mean. The upper and lower whisker indicate 95[th] and 5[th] percentiles, upper boundary, median line, and lower boundary of the box indicate the 75[th], 50[th], 25[th] quantiles, respectively. Data are from the NASA-CERES dataset.

[Figure]

Fig. R2 Decompositions of the mean seasonal variation of $R_{ld}$ ($\Delta$, difference between the maximum and minimum monthly data at each grid) in the NASA-CERES dataset into contributions by lower-level atmospheric heat storage ($\Delta R_{ld,T}$) (a) and emissivity ($\Delta R_{ld,\varepsilon}$) ($b$), and their latitudinal variations (c). Decomposed of $\Delta R_{ld,\varepsilon}$ into contributions by variations in cloud cover ($\Delta R_{ld,f_c}$) (d) and humidity ($\Delta R_{ld,e_a}$) (e), their latitudinal variations (f). In Figs. a, b, d, e, grey shading indicates missing values. In Figs. c and f, the box shows the variation among the land grids at the same latitude, while the solid line is their mean. The upper and lower whisker indicate 95th and 5th percentiles, upper boundary, median line, and lower boundary of the box indicate the 75th, 50th, 25th quantiles, respectively.

**Comments**:

1. Figs 1 and 5: This is totally a personal preference, but I don't like diverging colorbars when all the values are positive. I would use a sequential color bar for these figures to contrast them with the bias/difference maps in the rest of the paper.

[Reply] Thank the reviewer for the useful comment. We agree that diverging colorbar may give a wrong interpretation of the figure.

[Action] Using sequential colorbar in Figs. R3 and R2, which are Figs. 1 and 5 in the revised text, respectively.

[Figure]

Figure R3. Spatial distribution of the climatological mean (a, c) and the seasonal amplitude (b, d) of downward longwave radiation and absorbed solar radiation at the surface respectively from the NASA-CERES dataset. The seasonal amplitude is calculated as the difference between the maximum and minimum monthly data.

**2. Line 33: Check if it's the dominant term (isn't surface OLR bigger?)**

[Reply] Although the surface Outgoing Longwave Radiation (OLR) is the largest term in the surface energy balance (averaging 396 W/m² globally), we have deliberately chosen not to emphasize it in our discussion. Our focus is on highlighting the energy input in the surface energy balance, primarily comprising downward longwave radiation and absorbed solar radiation.

[Action] Revise the sentence as "*In the global mean surface energy budget, downward longwave radiation ($R_{ld}$) is the dominant surface energy input (333 $W/m^2$), contributing more than twice as much energy as absorbed solar radiation (161 $W/m^2$) (Trenberth et al. 2009)*".

**3. Line 44: This is a nice description of the development of Brutsaert's Equation, but I'd love for a bit more physical intuition of why we should expect the $1/T\_a$ dependence of the emissivity on temperature.**

[Reply] Thank you for the comment. The $1/T_a$ dependence of the emissivity on the temperature in equation 1 largely arises from the way Brutsaert 1975 derives the equation following the assumption of a standard atmosphere. Brutsaert derived the emissivity by integrating

Schwarzschild's equation (equation 2 in B75: $(F_{LU}, F_{LD}) = \int_{a=0}^{\infty} \pi B(T) \frac{\partial \epsilon_0(1.66a,T)}{\partial a} da$)).

According to that equation, the emissivity will depend on the scaled amount of water matter ($a$) for a clear sky. The scaled amount of water vapor ($a$) then in turn depends on humidity profiles (equations 3 and 8 in B75: $da = \rho_\omega (p/p_s)^{1/2} dz$ and $\rho_\omega = 0.622 \left( \frac{e_a}{R_d T_a} \right) \exp(-k_\omega z)$).

Hence, the term $\frac{e_a}{T_a} dz$ (or $e_a \frac{dz}{T_a}$) is introduced into the Equation, which essentially represents water vapor amount ($e_a$) considering the air volume increases ($T_a$) with temperature ($dz$). As we mentioned, Brutsaert then assumed a standard atmosphere and derived simple expressions for humidity profiles that solely depend on surface temperature ($T_a$) and surface water pressure ($e_a$) (equations 5, 7, and 8 in B75), while the integrated terms related to air volume ($dz$) is replaced by the empirical parameters "1.24" and "1/7".

However, we would like to point out that emissivity is largely insensitive to changes in $\frac{1}{T_a}$ (as demonstrated in the yellow line in Fig. R2f), which has also been acknowledged by B75. In fact, B75 provides another expression of emissivity that solely relies on water-vapor pressure at the surface (Equation 11' in B75: $\epsilon_{a0} = 0.553 \, e_a^{1/7}$). Moreover, we note that the dependence of $\frac{1}{T_a}$ on emissivity is also hidden in equation 1 due to changes in saturation vapor pressure with temperature. Therefore, it is hard to give a physical description of $\frac{1}{T_a}$ in Eq. 1 directly.

On the other hand, another dependence on emissivity on $T_a$ with physical explanations has been discussed in the current work. Under the freezing point, the emissivity will increase along with decreased temperature (Aase and Idso, 1978), which was not taken into account in B75 and partially contributes to the negative bias observed at low temperatures (Fig. R4a, which is Figure 3 in the revised manuscript).

[Actions]

(1) Following Eq.1, we add a sentence to make the point clear as "*Note that the dependence of $\varepsilon_{cs}$ on $1/T_a$ shown in Eq.1 should not be overinterpreted considering the derivation of the formula and also the correlation between $T_a$ and $e_a$.*"

(2) In Figure R2f, we have directly demonstrated the contribution of temperature anomalies to variations in emissivity. However, this contribution is of a smaller magnitude compared to that of cloud and water vapor, and therefore it has received less focus in our work.

(3) We have discussed to effect of temperature on emissivity under the freezing temperature to understand the underestimation of B75 at low temperatures (Fig. R4a) as follows.

[Figure]

Fig. R4 Biases in the estimates for multi-year mean $R_{ld}$ for FLUXNET data of 189 sites against air temperature (a) and water vapor pressure (b). Distribution of biases in the estimates for multi-year mean $R_{ld}$ for (c, d) NASA-CERES data from 2001 to 2018 and (e, f) ERA reanalysis from 1979 to 2021 for (c, e) clear-sky and (d, f) all-sky conditions over land. Grey shading indicates missing values.

4. Lines 50-51: I understand this equation is empirical, so the units don't have to make sense, but I think you should name the correct units for this equation in your description so readers who want to use this formula immediately know how to apply it. I also assume e_a corresponds to the 2m vapor pressure, but it's not stated explicitly.

[Reply] Thank you for pointing this out.

[Action] We have specified the unit of $e_a$ (millibars) and $T_a$ (K), as well as the definition of $e_a$.

5. Line 62: Can you be specific about what "this" estimate refers to, Is it Eq. 4, or 5?

[Reply] Thanks for pointing out a potential unclarity. "This" refers to Eqs. 4, and 5.

[Action] The sentence is rewritten for better clarity.

6. Lines 105-107: Is the cloud cover fraction from CERES used for all the calculations? If so, what's the relationship between cloud cover calculated from CERES and the very specific way it is defined in Eq. 3?

[Reply] We sincerely appreciate the reviewer for providing insightful and critical feedback. As shown in Fig. R5, the value derived from Equation 3 generally yields smaller cloud cover fractions compared to those obtained from both the CERES and ERA5 datasets. As a result, when estimating $R_{ld}$ based on Equation 3 under all sky conditions, the error distribution closely resembles that of clear sky conditions (Figures R4c-R4f). Moreover, this discrepancy affects the decomposition of seasonal variations, highlighting a higher contribution of water vapor anomalies to emissivity in tropical regions, particularly the Intertropical Convergence Zone, while cloud cover variations exert a greater influence over high latitudes (Figs. R2).

[Action] We have conducted all our analyses using fc calculated from Equation 3 instead of the cloud cover fraction.

[Figure]

Fig. R5 The scatter plot using the calculated cloud cover fraction (fc'=1-R/Rpot) and cloud cover from CERES datasets (fc). The dashed orange line is the linear regression while the solid blue line delicates the slope of 1.

7. Lines 127-131: Similar to my question about line 44, there's not much physical explanation for these biases, the authors merely note that Brutsaert also discussed how the equation was

biased for places with big temperature departures away from the global mean. I'm also not sure why the fact that the biases are less than the seasonal cycle is relevant.

[Reply] As mentioned in our response to comment #3, the underestimation observed at low temperatures, particularly near and below the freezing point, in Brutsaert's equation is attributed to its failure to account for the increase in Rld with decreasing temperatures below 0 ℃, as discussed in Aase and Idso (1978). It is worth noting that the biases identified, despite their presence, are smaller in magnitude compared to seasonal variations. Consequently, these biases do not hinder our utilization of the Brutsaert equation to attribute the causes behind seasonal variations.

[Action] In order to further enhance our understanding of the errors in B75, we have included this physical explanation in our work as follows: "*Moreover, B75 has not considered the gradual increase in emissivity as temperature decreases below freezing (Aase and Idso 1978), thus explaining the underestimation under low temperature (Figs. 3b, S3b, S3b). The biases seen in Figure 3 are nevertheless notably smaller than the spatial-temporal variations shown in Figure 1, this means that these biases do not prevent us from using Brutsaert to attribute the causes for the seasonal variation and the spatial range of $R_{ld}$.*"

8. Lines 132-135: Could the consistent positive bias in the all-sky calculation be driven by the cloud fraction definition related to my comment on lines 105-107? I think the definition of cloud fraction in these equations is pretty important to potential biases.

[Reply] Yes, the consistent positive bias in the all-sky calculation was driven by the way cloud fraction was defined from CERES. After repeating the analysis using a new metric of cloud-fraction as described in C&D 99, we found that the consistent positive bias in the all-sky calculation cannot be detected anymore (as shown in Fig. R4). We thank the reviewer for pointing this out.

9. Fig 3: Can you also show the biases from the AMERIFLUX dataset? Do those biases line up with the expectations from the global datasets?

[Reply] Thank you for this comment, which helps to justify our related statement.

[Action] Since the FLUXNET stations are primarily located in America and Europe, interpreting errors on the map figure becomes challenging. To address this, we have directly showcased the biases derived from the FLUXNET dataset alongside temperature and water vapor pressure (Figs. R4a-R4b), which demonstrates consistent patterns with the CERES and ERA5 datasets (Fig. R6, which is Fig. S3 in the supplementary material).

[Figure]

Figure R6. The same as Figs. R4a and R4b but with data from ERA5 (a and c) and CERES data (b and d)

10. Line 147: Is it small compared to     or just small compared to     from the Stephan Boltzman law? Are the water vapor and temperature components from the the emissivity equation of the same order of magnitude? If so, I think both temperature terms should be included.

[Reply] Although the pictures of the two equations in the comment are not available from our side, most likely due to a website issue, we assume that the reviewer is interested in understanding to which extent temperature variation contributes to the change in emissivity $(\Delta R_{ld,T_a'} = \sigma \overline{T_a}^4 \times (-\frac{1.24}{7}) \times \frac{(1-\overline{f_c})(\overline{e_a})^{\frac{1}{7}}}{(\overline{T_a})^{\frac{8}{7}}} \times \Delta T_a)$. This term is of less order than those of cloud $(\Delta R_{ld,f_c} = \sigma \overline{T_a}^4 \times \left(1 - 1.24 \left(\frac{e_a}{T_a}\right)^{\frac{1}{7}}\right) \times \Delta f_c)$ and water vapor $(\Delta R_{ld,e_a} = \sigma \overline{T_a}^4 \times \frac{1.24}{7} \frac{(1-\overline{f_c})}{(\overline{e_a})^{\frac{6}{7}}(\overline{T_a})^{\frac{1}{7}}} \times \Delta e_a)$, which are in turn of less order than the contribution of atmospheric storage $(\Delta R_{ld,T_a} = \varepsilon_a \sigma \overline{T_a}^3 \times \Delta T_a)$.

[Action] in the revision, we have substantiated this statement with Fig. R2f.

11. Lines 166-169: The monsoonal regions really stick out as having a sizable water vapor control on DLR. Even though the cloud cover causes changes in seasonal cycle strength, the

ultimate cause is changing water vapor or monsoonal circulation patterns, and I think this should be noted.

[Reply] Considering the substitution of the cloud cover fraction with the value calculated from Eq.3, the decomposition analysis has been impacted.

[Action] Consequently, we have conducted a reanalysis of the relevant data, revealing a noteworthy influence of water vapor on downward longwave radiation (DLR) over the Intertropical Convergence Zone (ITCZ) and monsoon regions (Fig. R2).

12. Lines 183-187: I'm confused about what's being plotted in Fig. 6a-d. Are these departures from the global mean that includes the ocean or just the terrestrial global mean?

[Reply] We apologize for not making it clear. It is the departure from the terrestrial global mean.

[Action] It has been made clearer in the caption as "**Figure 6.** *The decomposition of the annual-mean spatial variations (Δ, departure from the terrestrial global mean) of in the NASA-CERES dataset into contributions by (a) atmospheric heat storage ($\Delta R_{ld,T}$) and (b) emissivity ($\Delta R_{ld,\varepsilon}$). The variations in $\Delta R_{ld,\varepsilon}$ are further decomposed in contributions by variations in (c) cloud cover ($\Delta R_{ld,f_c}$) and (d) humidity ($\Delta R_{ld,e_a}$).*"

13. Line 195: 20W/m2 across the entire aridity index spectrum.

[Reply] Thank you for this detailed comment.

[Action] We have incorporated it into the text.

**Reference:**
Aase, J. K., and S. B. Idso, 1978: A comparison of two formula types for calculating long-wave radiation from the atmosphere. Water Resources Research, 14, 623-625. https://doi.org/10.1029/WR014i004p00623

Shakespeare C. J. and M. Roderick. (2022). Diagnosing Instantaneous Forcing and Feedbacks of Downwelling Longwave Radiation at the Surface: A Simple Methodology and Its Application to CMIP5 Models. Journal of Climate.

Vargas Zeppetello, L. R., Donohoe, A., & Battisti, D. S. (2019). Does surface temperature respond to or determine downwelling longwave radiation? Geophysical Research Letters, 46, 2781–2789. https://doi.org/10.1029/2019GL082220